# Health Behavior Change and Complementary Medicine Use: National Health Interview Survey 2012

**DOI:** 10.3390/medicina55100632

**Published:** 2019-09-24

**Authors:** Felicity L Bishop, Romy Lauche, Holger Cramer, Jonquil W Pinto, Brenda Leung, Helen Hall, Matthew Leach, Vincent CH Chung, Tobias Sundberg, Yan Zhang, Amie Steel, Lesley Ward, David Sibbritt, Jon Adams

**Affiliations:** 1Department of Psychology, University of Southampton, Southampton SO17 1BJ, UK; J.W.Pinto@soton.ac.uk; 2Australian Research Centre in Complementary and Integrative Medicine (ARCCIM), University of Technology Sydney, Sydney PO Box 123, Broadway NSW 2007, Australia; Romy.Lauche@uts.edu.au (R.L.); h.cramer@kem-med.com (H.C.); brenda.leung@uleth.ca (B.L.); helen.hall@monash.edu (H.H.); matthew.leach@unisa.edu.au (M.L.); tobias.sundberg@ki.se (T.S.); YAN.ZHANG@tcu.edu (Y.Z.); Amie.Steel@uts.edu.au (A.S.); lesley.ward@northumbria.ac.uk (L.W.); David.Sibbritt@uts.edu.au (D.S.); Jon.Adams@uts.edu.au (J.A.); 3Sozialstiftung Bamberg, 96049 Bamberg, Germany; 4Department of Internal and Integrative Medicine, University of Duisburg-Essen, 45276 Essen, Germany; 5Faculty of Health Sciences, University of Lethbridge, Alberta, T1K 3M4, Canada; 6Faculty of Medicine, Monash University, Clayton, VIC 3800, Australia; 7Department of Rural Health, University of South Australia, Whyalla Norrie SA 5608, Australia; 8Jockey Club School of Public Health & Primary Care and School of Chinese Medicine, The Chinese University of Hong Kong, Sha Tin 999077, Hong Kong; 9Musculoskeletal & Sports Injury Epidemiology Center (MUSIC), Institute of Environmental Medicine, Karolinska Institutet, 171 77 Stockholm, Sweden; 10Harris College of Nursing and Health Sciences, Texas Christian University, Fort Worth, TX 76109, USA; 11Department of Sport, Exercise & Rehabilitation, Northumbria University, Newcastle-upon-Tyne NE1 8ST, UK

**Keywords:** health behavior, complementary and alternative medicine, health attitudes, motivations, lifestyle

## Abstract

*Background**and**objectives*: Complementary and alternative medicine (CAM) use has been associated with preventive health behaviors. However, the role of CAM use in patients’ health behaviors remains unclear. This study aimed to determine the extent to which patients report that CAM use motivates them to make changes to their health behaviors. *Materials*
*and*
*Methods*: This secondary analysis of 2012 National Health Interview Survey data involved 10,201 CAM users living in the United States who identified up to three CAM therapies most important to their health. Analyses assessed the extent to which participants reported that their CAM use motivated positive health behavior changes, specifically: eating healthier, eating more organic foods, cutting back/stopping drinking alcohol, cutting back/quitting smoking cigarettes, and/or exercising more regularly. *Results*: Overall, 45.4% of CAM users reported being motivated by CAM to make positive health behavior changes, including exercising more regularly (34.9%), eating healthier (31.4%), eating more organic foods (17.2%), reducing/stopping smoking (16.6% of smokers), or reducing/stopping drinking alcohol (8.7% of drinkers). Individual CAM therapies motivated positive health behavior changes in 22% (massage) to 81% (special diets) of users. People were more likely to report being motivated to change health behaviors if they were: aged 18–64 compared to those aged over 65 years; of female gender; not in a relationship; of Hispanic or Black ethnicity, compared to White; reporting at least college education, compared to people with less than high school education; without health insurance. *Conclusions*: A sizeable proportion of respondents were motivated by their CAM use to undertake health behavior changes. CAM practices and practitioners could help improve patients’ health behavior and have potentially significant implications for public health and preventive medicine initiatives; this warrants further research attention.

## 1. Introduction

Health behaviors such as physical activity, diet, and smoking have a significant impact on mortality and morbidity worldwide [1]. Consultations with health providers—especially primary health care professionals—offer an opportunity to initiate health behavior change, although interventions to motivate such change have had mixed results to date. For example, one cluster-randomized trial found General Practitioner (GP)-delivered advice effectively increased physical activity levels among sedentary adults [2]. Yet, in another randomized trial, GP-delivered behavioral counselling failed to reduce levels of four risky health behaviors (smoking, alcohol use, exercise, and healthy eating) among primary care patients [3]. Patients may find behavior change programs more acceptable when they perceive them as patient-centered, and thus such program designs may be more effective than others at changing health behaviors [4], but evidence on this point is heterogeneous [5]. Another strand of research is focused on the identification and evaluation of effective components of interventions to improve health behaviors, known as behavior change techniques [6]. Some techniques, such as goal-setting, have shown promise [7,8], but the complexities of which techniques work for which patients, in what circumstances, and why, are still being investigated [9]. Given the mixed success of integrating health behavior change interventions within the context of mainstream medical care, it is important to consider whether and how other settings and providers might contribute to improving public health behaviors.

Complementary and alternative medicine (CAM) refers to a broad collection of self-care and practitioner-based practices that have a history of use outside of conventional healthcare. According to the National Center for Complementary and Integrative Health (NCCIH), the most popular CAM therapies usually fall under one of two major categories: natural products (such as herbal medicine and dietary supplementation), and mind-body practices (such as acupuncture, massage, meditation, and yoga) [10]. Many CAM treatments are practiced not in isolation but as part of a philosophically-driven system of care, which integrates a range of treatments. Traditional Chinese Medicine (TCM), for example, commonly encompasses herbal medicines, acupuncture, tai chi, and/or qi gong. Lifestyle advice is often provided during consultations as an essential part of healing in line with TCM philosophy [11]. Mind-body practices have also been specifically highlighted as providing an opportunity to address patients’ health behaviors [12].

A substantial number of people around the world use CAM, with reports ranging from 10% to 76% of the general population having used CAM within the past 12 months [13]. People use CAM for diverse reasons including to treat or manage (the symptoms of) disease and for wellness in the presence or absence of chronic conditions [14,15]. CAM use has previously been shown to be correlated with engagement in more positive health behaviors and fewer risky health behaviors [16], while adults who use CAM for health promotion may be especially likely to engage in healthy behaviors (compared to those who use CAM therapeutically) [15]. However, a significant proportion of CAM users present with health risk factors, including obesity and physical inactivity, suggesting an opportunity for CAM practitioners to engage in health behavior change work [17]. Furthermore, patients of CAM practitioners report empathetic, individualized, or patient-centered consultation experiences [18], which may make these practitioners more likely to facilitate successful behavior change via patient empowerment. Small scale qualitative work in specific CAM therapies such as acupuncture suggests some practitioners may offer self-care advice that explicitly includes health behavior change [19]. However, it is not yet known whether such practices are idiosyncratic or more widespread, or the extent to which patients themselves feel motivated by their CAM use to make health behavior changes. It is important to address these questions in order to help maximize the potential of CAM practitioners to possibly influence wider public and preventive health advances and act as important preventive medicine advocates and health behavior advisors within the broader health care system. Indeed, when considered in the context of the high levels of CAM use [20], the need for further examination of this issue is compelling.

In direct response to these circumstances, this paper reports the findings from an analysis of 2012 National Health Interview Survey (NHIS) data with the aim of determining the proportion of CAM users who are motivated by their CAM use to change their health behaviors. The objectives were to determine (1) what proportion of people using different types of CAM report being motivated by their CAM use to change their health behaviors, (2) which health behaviors do people report changing as a consequence of their CAM use and (3) which sociodemographic and health characteristics are associated with being motivated by CAM use to change one’s health behaviors. Findings from this work may shed light on potentially significant opportunities for facilitating successful health behavior change at the individual, clinical, and population level.

## 2. Materials and Methods

### 2.1. Design and Participants

This secondary analysis used data extracted from the 2012 National Health Interview Survey (NHIS)—a periodically conducted cross-sectional household interview survey targeting the non-institutionalized civilian population of the United States. The NHIS obtains free and informed consent from participants. The current analysis was approved by the lead author’s institution and utilized the 2012 NHIS Family Core (i.e., Demographic and Health Characteristics), and the Adult Alternative Medicine supplement (i.e., CAM use and related variables including behavior changes). The total household response rate for the 2012 NHIS was 77.6%. The interviewed sample consisted of 42,366 eligible households, which yielded 34,525 respondents aged 18 years and older. Further details of the NHIS sample are reported elsewhere [21].

Respondents to the 2012 NHIS survey were asked to specify, “During the past 12 months, which three CAM therapies were the most important for your health?” The question did not specify whether these therapies should be practitioner-based or self-directed/self-help practices, thus both may have been reported. The response options were: chiropractic or osteopathic manipulation; massage; acupuncture; energy healing therapy; naturopathy; hypnosis; biofeedback; craniosacral therapy; traditional healers; herbs; homeopathy; meditation; yoga/tai chi/qi gong; special diets; and movement or exercise therapies. From the respondents, 5487 specified one CAM therapy, 2638 specified two CAM therapies, and 2076 specified three CAM therapies as most important to their health. Overall, 1058 respondents reported using more than three CAM therapies but these individuals were also restricted to specifying up to 3 therapies that they deemed most important for their health. The respondents were not asked to select their top therapies in order; therefore, it is important to analyze data related to all the top-ranked therapies as specified by respondents. Henceforth, the 10,201 adult respondents who reported one or more top-ranked CAM therapies were included in the sample analyzed and reported in this paper.

### 2.2. Measures

To measure CAM-motivated health behavior change, respondents were asked five questions about health behavior change in relation to their top-ranked CAM(s). These questions explicitly asked whether or not having consulted/employed their top-ranked CAM practitioner/self-care modality in the past 12 months motivated the respondent to: (1) cut back or stop drinking alcohol, (2) cut back or stop smoking cigarettes, (3) eat healthier, (4) eat more organic foods, and (5) exercise more regularly.

The following demographic and health characteristics were examined: age, gender, ethnicity, region, education, marital status, body mass index, self-rated health status, health insurance coverage, and number of chronic conditions (defined as having been diagnosed with any of the following conditions inquired about in the survey: hypertension, coronary heart disease, stroke, asthma, cancer, diabetes, kidney problems, arthritis, hepatitis, or chronic obstructive pulmonary disease [COPD]).

### 2.3. Statistical Analysis

SPSS (version 24; IBM Corp, Armonk, NY, USA) was used for all analyses. Data were weighted based on the sample size adjusted weight for the US population to take account of the complex survey design [21]. At most, 1.6% of the 10,201 cases had missing data (18 on education, 2 employment, 1 health status, 87 insurance); these were handled via pairwise deletion.

To compute the overall prevalence of any health behavior changes, respondents were categorized according to whether or not they had been motivated to make each of the 5 behavior changes (cut back or stop drinking alcohol; cut back or stop smoking cigarettes; eat healthier; eat more organic foods; exercise more regularly) after using any of their top-ranked CAMs. To compute the prevalence of health behavior changes following use of specific CAM modalities, we summed the number of people making each health behavior change after each specific CAM modality, regardless of whether that modality was the first, second, or third top-ranked CAM as specified by a respondent.

Cross tabs were used to compare rates of health behavior change by top CAM therapy. Pearson’s chi-squared was computed to compare health behavior change rates in groups with different demographic and health characteristics. Logistic regression analysis was used to identify demographic and health characteristics that were independent predictors of making a CAM-motivated health behavior change; variables were forced into the model and odds ratios with 95% confidence intervals were calculated. Due to the large sample size, statistical significance was set at *p* < 0.005.

### 2.4. Ethics

Ethics approval for this secondary data analysis was obtained from the University of Southampton Psychology Ethics Committee (Submission Number: 26888).

## 3. Results

### 3.1. Prevalence of Health Behavior Changes

Overall, 45.4% of respondents reported being motivated to make at least one health behavior change after using CAM. Approximately, one third of respondents reported being motivated to exercise more regularly (34.9%) or to eat more healthily (31.4%); 17.2% of respondents reported being motivated to eat more organic foods; 16.6% of smokers reported being motivated to cut back or stop smoking cigarettes; and 8.7% of alcohol drinkers reported being motivated to cut back or stop drinking alcohol. Individual CAM modalities all inspired health behavior change in some users (see Table 1). Those CAM modalities that, by definition, involve certain integral and explicit health behavior changes were most likely to motivate respondents to change the behaviors integral to that modality. For example, 66.7% of people using movement or exercise techniques and 61.5% of those using yoga or tai chi/qi gong reported being motivated to exercise more regularly, while 77.3% of those following a special diet reported being motivated to eat more healthily. However, these therapies also motivated other behavior changes: 43.4% of people using movement or exercise techniques and 41.2% of those using yoga or tai chi/qi gong reported being motivated to eat more healthily; and 39.5% of those following a special diet reported being motivated to exercise more regularly.

### 3.2. Characteristics of CAM Users Motivated to Change Health Behaviors

Table 2 shows the distribution of demographic and health characteristics of CAM users by motivation to change at least one health behavior. Statistically significant associations were observed between motivation to change after using CAM and all of the demographic and health characteristics.

The magnitudes of these associations were quantified using a logistic regression model (see Table 3). The multiple regression model showed that respondents with the following characteristics were significantly more likely to be motivated by CAM use to change at least one health behavior: 18–64 year olds (ORs = 1.64 to 2.13; compared to those aged over 65 years); females (OR = 1.61); people not in a relationship (OR = 1.24); people of Hispanic or Black ethnicity (ORs = 1.27 and 1.45, compared to White); people with at least college education (OR = 1.53; compared to people with less than high school education); and people with no health insurance (OR = 1.38; compared to those with private health insurance). People with 2 health conditions were less likely to change health behavior than those with no conditions (OR = 0.80). In this multivariable model, geographical region, body mass index, health status, and other numbers of chronic conditions were not significantly associated with being motivated by CAM use to change at least one health behavior.

## 4. Discussion

This analysis of 2012 NHIS data shows more than 45% of CAM users report being motivated by their CAM use to make positive health behavior changes, most notably in the areas of exercise and diet, but also in reducing smoking and alcohol intake. These positive changes occurred following the use of both natural products and mind-body practices.

Survey respondents self-reported being motivated by their CAM use to change health behaviors. However, as causation cannot be determined in this cross-sectional self-reported survey, it remains unclear whether CAM use directly and independently motivates behavior change, or whether being predisposed to make health behavior changes drives the choice to use CAM. On the one hand, advice for health behavior change and self-management are integral parts of many CAM interventions and are often driven by the philosophical underpinnings of whole systems of care [11,22]. The collaborative and patient-centered style that characterizes some CAM encounters [23,24] may also foster autonomy and empowerment, thus potentially motivating health behavior change [18,25,26]. Lending support to this interpretation, positive health behavior changes including increased physical activity, stress management, and dietary changes have been observed in chronically ill patients *after* participation in inpatient [27] and outpatient [28] CAM-based lifestyle modification programs. Conversely, an older study from the UK suggests that patients consulting CAM practitioners are often more interested in adopting a healthy lifestyle than are patients solely consulting general practitioners [29]. Indeed, cancer patients using CAM therapies have been shown to be much more likely to additionally use conventional lifestyle therapies such as dietary changes, conventional supplements, and exercise than patients not using CAM [30]. Furthermore, many CAM users consider the lifestyle guidance offered within these systems of care as a key reason for choosing to use such modalities [31]. Thus, there appears to be a bidirectional pathway at play with CAM practitioners being more likely than conventional practitioners to include health behavior advice as part of their treatment, and CAM users already being more interested in health behavior changes than CAM non-users.

Initiating and maintaining health behavior changes requires a specific set of beliefs and expectancies, some of which may overlap with beliefs fostered by CAM use. Evidence suggests that, to successfully implement health behavior changes, it is particularly important for patients to have high self-efficacy expectations (i.e., to believe in one’s ability to make a specific behavior change) [32,33]. Having an internal health locus of control (i.e., a general belief that one’s actions can influence one’s own health) is also positively associated with undertaking health behavior change [27,34]. CAM use has been shown to shift an individual’s beliefs about control of health outcomes towards a more internal and less external locus of control [35]. Likewise, participating in mind-body practices can increase exercise self-efficacy [36]. Thus, the motivational effect of CAM use for initiating positive health behavior change may be explained at least partly by an increase in health behavior-related self-efficacy as well as a change in health locus of control as a consequence of CAM use.

In line with previous research in other populations, our analyses have shown that among CAM users, being motivated to change health behaviors was most strongly associated with younger age [37], female sex [38], and higher levels of education [38]. Our analyses also showed that uninsured CAM users were more likely to make health behavior changes. Associations between health insurance coverage and health behavior change have not been thoroughly explored previously, although lack of health insurance may impact on an individual’s access to health services [39] and encourage such individuals to undertake preventive self-care practices to mitigate out-of-pocket health care costs. This view is consistent with evidence showing uninsured people are more likely to use CAM compared to those with public or private health insurance [16].

The strengths of this research include the use of a nationally representative dataset, a large sample size, and the examination of the motivational effects of specific, individual CAM modalities upon the health behavior change of users. However, the secondary analysis of self-reported data poses five main limitations. One, respondents were asked to recall the details of their CAM use over the previous 12 months; as such, the data may be subject to recall bias. Two, respondents were only asked about motivation for health behavior change with regards to their three top-ranked CAM therapies; the potential influence of additional modalities and consumption was not explored and this may have led to an underestimation of the motivational impact of CAM use in our analyses. Three, respondents were only asked about their motivation for health behavior change, not whether they actually initiated and, perhaps even more important, maintained health behavior changes in their daily lives—an area that warrants further investigation. Four, the survey questions probing the respondents’ top-ranked CAM therapies did not distinguish between practitioner-directed and self-directed CAM use and future studies on health behavior change and CAM should make this distinction to help facilitate a richer understanding of the mechanisms involved. Five, findings may not generalize to other countries with different healthcare systems.

The findings of this analysis have potential implications for future research and initiatives in preventive behavioral medicine and public health. Future research should investigate, amongst other topics: which means of administering CAM (e.g., practitioner-directed CAM versus patient-directed CAM self-care) and which specific CAM modalities are best-suited to helping people sustain positive health behavior changes; how CAM use and health behavior change are inter-related over time; the detailed ways in which CAM practitioners and practices encourage health behavior change including, for example, the role of practitioner support and increasing patients’ sense of responsibility for their health [40]; and the potential for CAM practitioners to successfully integrate positive behavior change management within a collaborative interdisciplinary approach to patient care alongside conventional providers and broader preventive medicine initiatives. In terms of implications for practice, a better understanding of how CAM practitioners facilitate health behavior change may offer fresh approaches to health psychologists, public health workers, and others designing and delivering health behavior change interventions—such as taking CAM diagnostic criteria into account [19] or expanding repertoires of techniques for health behavior change [6]. For those CAM practitioners recommending health behavior changes on a more opportunistic and/or idiosyncratic basis, health psychologists are well-placed to provide any necessary training [17] to help these practitioners facilitate health behavior change (where applicable) amongst *all* their patients.

## 5. Conclusions

In conclusion, a sizeable proportion of CAM users (>45%) are motivated by their CAM use to make health behavior changes. Changes to exercise and diet are most commonly reported, but CAM users also report being motivated to reduce their smoking and alcohol intake. CAM users who are younger, female, more highly educated, and uninsured are more likely to make health behavior changes. The use of natural products and mind-body practices motivates people to improve their health behaviors. This phenomenon warrants further investigation and presents an opportunity for interdisciplinary collaboration between CAM and health psychology, public health, and preventive medicine.

## Figures and Tables

**Table 1 medicina-55-00632-t001:** Prevalence of Health Behavior Changes Motivated by Using Individual Complementary and Alternative Medicine (CAM) Modalities (n = 10,201).

Top Therapy	n	Any Health Behavior Change	Eat Healthier	Eat More Organic Foods	Reduce Alcohol Intake ^1^	Reduce Smoking ^2^	Exercise More Regularly
Special diets	898	81.0%	77.3%	46.6%	20.4%	17.2%	39.6%
Movement or exercise therapies	593	70.2%	43.4%	18.4%	11.0%	33.3%	66.7%
Naturopathy	74	67.6%	62.2%	39.2%	12.2%	36.4%	35.1%
Yoga/Tai chi/qi gong	2698	67.4%	41.2%	22.2%	11.9%	25.7%	61.5%
Meditation	1338	39.6%	28.5%	20.0%	13.4%	27.8%	29.5%
Hypnosis	66	37.9%	24.2%	10.6%	9.6%	33.3%	24.2%
Traditional healers	115	37.7%	28.4%	19.0%	16.4%	7.8%	19.1%
Energy healing therapy	80	37.5%	28.8%	16.0%	6.1%	25.0%	27.2%
Homeopathy	504	33.9%	29.6%	21.5%	9.7%	20.6%	19.3%
Biofeedback	77	31.2%	15.6%	9.1%	1.9%	12.5%	20.8%
Craniosacral therapy	41	29.3%	19.5%	17.1%	10.7%	10.0%	22.0%
Chiropractic or osteopathic manipulation	2710	25.6%	10.5%	5.7%	2.6%	6.6%	21.4%
Acupuncture	418	25.2%	15.0%	11.7%	6.5%	17.2%	17.5%
Herbs	5373	24.7%	18.0%	9.4%	3.7%	6.2%	13.6%
Massage	2005	22.5%	11.5%	6.9%	2.8%	3.8%	18.8%

^1^ Only people who had previously reported drinking alcohol were queried about whether CAM motivated them to reduce their alcohol intake. This number varies by therapy as follows: special diets (n = 618), movement or exercise techniques (n = 500), naturopathy (n = 49), yoga/Tai chi/qi gong (n = 2101), meditation (n = 1025), hypnosis (n = 52), traditional healers (n = 73), energy healing therapy (n = 49), homeopathy (n = 359), biofeedback (n = 53), craniosacral therapy (n = 28), Chiropractic or osteopathic manipulation (n = 2006), acupuncture (n = 291), herbs (n = 3924), massage (n = 1585). ^2^ Only people who had previously reported smoking cigarettes were queried about whether CAM motivated them to reduce their smoking. This number varies by therapy as follows: special diets (n = 344), movement or exercise techniques (n = 27), naturopathy (n = 11), yoga/Tai chi/qi gong (n = 319), meditation (n = 245), hypnosis (n = 15), traditional healers (n = 51), energy healing therapy (n = 16), homeopathy (n = 136), biofeedback (n = 16), craniosacral therapy (n = 20), Chiropractic or osteopathic manipulation (n = 366), acupuncture (n = 58), herbs (n = 700), massage (n = 238).

**Table 2 medicina-55-00632-t002:** Characteristics of CAM users and Motivation to Change Health Behaviors.

Characteristic	Not Motivated to Change Health Behavior		Motivated to Change at Least One Health Behavior		
	n	%	n	%	*p*
Age (years)					<0.001 *
18–29	814	43.8%	1042	56.2%	
30-39	934	50.6%	910	49.4%	
40-49	1052	55.5%	842	44.5%	
50–64	1666	56.0%	1312	44.0%	
65 plus	1107	67.9%	523	32.1%	
Gender					<0.001 *
Male	2548	61.0%	1627	39.0%	
Female	3024	50.2%	3002	49.8%	
Marital status					<0.001 *
Not in relationship	1820	50.0%	1823	50.0%	
In relationship	3752	57.2%	2806	42.8%	
Ethnicity					<0.001 *
White	4473	56.6%	3425	43.4%	
Hispanic	473	49.0%	493	51.0%	
Black	318	45.8%	376	54.2%	
Asian	280	49.4%	287	50.6%	
Other	29	37.8%	48	62.2%	
Region					0.003 *
West	1520	52.2%	1393	47.8%	
Northeast	980	56.5%	755	43.5%	
Midwest	1476	56.6%	1133	43.4%	
South	1597	54.2%	1348	45.8%	
Education					<0.001 *
Less than high school	386	61.2%	245	38.8%	
High school	2390	57.5%	1768	42.5%	
College or higher	2780	51.7%	2595	48.3%	
Health insurance coverage					<0.001 *
Uninsured	568	46.1%	664	53.9%	
At least public health insurance	792	58.6%	558	41.4%	
Private health insurance	4122	55.4%	3321	44.6%	
Body Mass Index (kg/m2)					<0.001 *
< 18.5	93	51.3%	88	48.7%	
18.5–25	1950	50.7%	1893	49.3%	
25–30	1935	56.5%	1488	43.5%	
>30	1594	57.9%	1159	42.1%	
Subjective health status					<0.001 *
Very good or excellent	3477	52.3%	3173	47.7%	
Good	1460	58.4%	1038	41.6%	
Fair or poor	634	60.3%	417	39.7%	
Number of chronic conditions					0.001 *
0	2744	50.4%	2705	49.6%	
1	1504	57.2%	1127	42.8%	
2	783	63.9%	443	36.1%	
3	318	62.4%	192	37.6%	
4 or more	201	61.2%	128	38.8%	

* *p* < 0.005.

**Table 3 medicina-55-00632-t003:** Adjusted Odds Ratios for Demographic and Health Variables Predicting Health Behavior Change Motivated by CAM.

Characteristic	Category	OR	Lower CI	Upper CI	*p*
Age (years)	65 plus	Reference			
	50–64	1.599	1.383	1.850	<0.001 *
	40–49	1.457	1.236	1.717	<0.001 *
	30–39	1.662	1.404	1.966	<0.001 *
	18–29	2.127	1.794	2.522	<0.001 *
Gender	Male	Reference			
	Female	1.612	1.479	1.757	<0.001 *
Marital status	In relationship	Reference			
	Not in relationship	1.237	1.132	1.353	<0.001 *
Ethnicity	White	Reference			
	Hispanic	1.273	1.097	1.477	0.001 *
	Black	1.447	1.225	1.710	<0.001 *
	Asian	1.154	0.962	1.385	0.122
	Other	1.909	1.174	3.103	0.009
Region	Midwest	Reference			
	Northeast	0.967	0.850	1.099	0.603
	West	1.132	1.011	1.267	0.032
	South	1.041	0.931	1.163	0.484
Education	Less than high school	Reference			
	High school	1.197	0.994	1.441	0.058
	College or higher	1.527	1.265	1.844	<0.001 *
Health insurance coverage	Private health insurance	Reference			
	At least public health insurance	1.194	1.041	1.369	0.011
	Uninsured	1.382	1.211	1.576	<0.001 *
Body mass index (kg/m^2^)	>30	Reference			
	25–30	1.120	1.003	1.250	0.044
	18.5–25	1.009	0.734	1.387	0.958
	<18.5	1.109	0.995	1.236	0.062
Self-rated health status	Fair or poor	Reference			
	Good	1.169	0.997	1.370	0.054
	Very good or excellent	1.024	0.870	1.205	0.777
Number of chronic conditions	0	Reference			
	1	0.897	0.809	0.995	0.040
	2	0.795	0.686	0.921	0.002 *
	3	0.912	0.737	1.129	0.398
	4 or more	1.162	0.894	1.510	0.261

* *p* < 0.005; Note. n = 9860. Model χ^2^ = 521.93, df = 27, *p* < 0.001.

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
