# Peer review of "Health Behavior Change and Complementary Medicine Use: National Health Interview Survey 2012"

_medicina, 2019, doi:10.3390/medicina55100632_

Round 1

Reviewer 1 Report

A concise and well-written manuscript. The authors are clear about the limitations of the study and the conclusions drawn are valid.

Could the introduction make reference to similar work based on the 2007 data (Hawk, 2012; Preventive Medicine) and explain how it builds on this? Although there is no explicit mention of behaviour change that study’s conclusion is somewhat similar.

Suggest findings in Table 3 be presented in a figure so that comparisons are more easily made between the demographic and health variables. (This is simply a recommendation, not a request).

Although it is true that the survey did not distinguish between practitioner-directed and self-directed CAM, there are some interesting trends in Table 1. If this was re-formatted to show Any health behaviour change in descending order it would more easily illustrate which CAM modalities inherently promote behaviour change and also which modalities could potentially have an effect irrespective of this.

The influence and level of practitioner support (e.g. Williams-Piehota, 2011; Alternative Therapies) should also be mentioned in the final paragraph of the Discussion given the emphasis on training CAM practitioners.

Author Response

We thank the reviewer for their constructive comments.

COMMENT:  Could the introduction make reference to similar work based on the 2007 data (Hawk, 2012; Preventive Medicine) and explain how it builds on this? Although there is no explicit mention of behaviour change that study’s conclusion is somewhat similar.

RESPONSE:  We have added reference to Hawk 2012 in the Introduction: However, a significant proportion of CAM users present with health risk factors, including obesity and physical inactivity, suggesting an opportunity for CAM practitioners to engage in health behavior change work [17].” 

COMMENT:  Suggest findings in Table 3 be presented in a figure so that comparisons are more easily made between the demographic and health variables. (This is simply a recommendation, not a request).

RESPONSE: We prefer to retain the Table as it provides more information than would be possible in a Figure.

COMMENT:  Although it is true that the survey did not distinguish between practitioner-directed and self-directed CAM, there are some interesting trends in Table 1. If this was re-formatted to show Any health behaviour change in descending order it would more easily illustrate which CAM modalities inherently promote behaviour change and also which modalities could potentially have an effect irrespective of this.

RESPONSE:  We appreciate this suggestion and have re-formatted Table 1 as suggested, in descending order of any health behaviour change.

COMMENT:  The influence and level of practitioner support (e.g. Williams-Piehota, 2011; Alternative Therapies) should also be mentioned in the final paragraph of the Discussion given the emphasis on training CAM practitioners.

RESPONSE:  We have added reference to Williams-Piehota (2011) as follows:  “The findings of this analysis have potential implications for future research and initiatives in preventive behavioral medicine and public health. Future research should investigate, amongst other topics: which means of administering CAM (e.g. practitioner-directed CAM versus patient-directed CAM self-care) and which specific CAM modalities are best-suited to helping people sustain positive health behavior changes; how CAM use and health behavior change are inter-related over time; the detailed ways in which CAM practitioners and practices encourage health behavior change including, for example, the role of practitioner support and increasing patients’ sense of responsibility for their health [40];…”

Reviewer 2 Report

Review of the manuscript "Health behavior change and complementary medicine use: National Health Interview Survey 2012"

To begin, I would like to congratulate the authors of the manuscript on encroaching on the topic related to changes in health behaviors under the influence of using complementary and alternative medicine (CAM) services. The combination of advice provided by medical staff and CAM therapy can change health behaviors into healthier ones. Indeed, the strength of the research is the large study sample, representative of the USA. However, I would like to advisesome minor corrections to organize the content of the manuscript, making it more accessiblefor the reader. These modifications should concern the following elements:

At the end of the Introduction section, authors should clearly define the purpose of the study. In the Materials and Methods section, the authors should distinguish subchapters, such as Study Design and Participants, the method of selecting study participants, respondents excluded from the study, etc., and not as it is in the current form included in the paragraph describing the methods of statistical analysis. The next subsection in this part should refer to the study tools used to carry out the tests. Moreover, the last one should be entitled Statistical Analysis. In my opinion, the conclusions should be revised so that they reflect the purpose of the research. The manuscript requires many editing corrections, as recommended by the journal.

To sum up, after making the above changes, I recommend accepting the manuscript for publication in the journal Medicina.

Author Response

We thank the reviewer for their constructive comments.

COMMENT:  At the end of the Introduction section, authors should clearly define the purpose of the study.

RESPONSE:  We have expanded our existing statement of the aims by adding “The objectives were to determine (1) what proportion of people using different types of CAM report being motivated by their CAM use to change their health behaviors, (2) which health behaviors do people report changing as a consequence of their CAM use and (3) which sociodemographic and health characteristics are associated with being motivated by CAM use to change one’s health behaviors.”

COMMENT:  In the Materials and Methods section, the authors should distinguish subchapters, such as Study Design and Participants, the method of selecting study participants, respondents excluded from the study, etc., and not as it is in the current form included in the paragraph describing the methods of statistical analysis.

The next subsection in this part should refer to the study tools used to carry out the tests.

Moreover, the last one should be entitled Statistical Analysis.

RESPONSE:  Thank you for the suggestions to incorporate subheadings into the Materials and Methods section.  We have made these changes. 

COMMENT:  In my opinion, the conclusions should be revised so that they reflect the purpose of the research.

RESPONSE:  We have expanded our conclusions to better reflect the objectives, as follows:  “In conclusion, a sizeable proportion of CAM users (>45%) are motivated by their CAM use to make health behavior changes. Changes to exercise and diet are most commonly reported, but CAM users also report being motivated to reduce their smoking and alcohol intake. CAM users who are younger, female, more highly educated, and uninsured are more likely to make health behavior changes. The use of natural products and mind-body practices motivates people to improve their health behaviors.”

COMMENT:  The manuscript requires many editing corrections, as recommended by the journal.

RESPONSE:  We are happy to make any further edits as requested by the journal.